# Exosome Shedding Is Concordant with Objective Treatment Response Rate and Stratifies Time to Progression in Treatment Naïve, Non-Resectable Hepatocellular Carcinoma

Kelley G. Núñez [1,†], Dorota Wyczechowska [2,†], Mina Hibino [1], Tyler Sandow [3], Juan Gimenez [3], Ali R. Koksal [4], Yucel Aydin [4], Srikanta Dash [4], Ari J. Cohen [5,6] and Paul T. Thevenot [1,*]

1. Institute of Translational Research, Ochsner Health System, New Orleans, LA 70121, USA; kelley.nunez@ochsner.org (K.G.N.); mina.hibino@ochsner.org (M.H.)
2. Department of Interdisciplinary Oncology, Stanley S. Scott Cancer Center, LSU-LCMC Cancer Center, Louisiana State University Health Science Center, New Orleans, LA 70112, USA; dwycze@lsuhsc.edu
3. Interventional Radiology, Ochsner Health System, New Orleans, LA 70121, USA; tyler.sandow@ochsner.org (T.S.); juan.gimenez@ochsner.org (J.G.)
4. Department of Pathology and Laboratory Medicine, Tulane University Health Sciences Center, New Orleans, LA 70112, USA; akoksal@tulane.edu (A.R.K.); yaydin@tulane.edu (Y.A.); sdash@tulane.edu (S.D.)
5. Multi-Organ Transplant Institute, Ochsner Health System, New Orleans, LA 70121, USA; acohen@ochsner.org
6. Faculty of Medicine, The University of Queensland, Brisbane 4072, Australia
* Correspondence: paul.thevenot@ochsner.org
† These authors contributed equally to this work.

**Abstract:** Translational strategies to characterize and monitor extracellular vesicles such as exosome (EX) shedding and the clinical impact of this data within hepatocellular carcinoma (HCC) remains unclear. In this study, EX shedding was assessed in early-stage HCC and evaluated as a stratification factor for time to progression (TTP) following first-cycle liver-directed therapy (LDT). Plasma EXs were isolated from HCC patients undergoing LDT using ultracentrifugation. Purified EXs were stained using markers CD9 and CD63 and quantified using an ImageStreamX flow cytometer. Circulating EXs expressing CD9 were isolated at 10-fold higher levels compared to CD63. The intensity of CD9+ EX shedding following LDT was positively correlated with treatment response. High post-LDT CD9+ EX shedding stratified TTP risk with a 30% lower frequency of disease progression at 1 year following LDT. Post-LDT high CD9+ EX shedding was observed in 100% (10/10) of patients successfully bridged to liver transplantation while only 22% (2/9) of patients with tumor progression had high CD9+ EX shedding post-LDT. CD9+ EX shedding also stratified TTP risk within the first cycle objective response rate (ORR) group, identifying patients still at higher disease progression. EX shedding was concordant with imaging response rate, stratified TTP in early-stage HCC, and may have important implications for assessing post-LDT viable, biologically aggressive HCC.

**Keywords:** exosomes; hepatocellular carcinoma; liver-directed therapy

## 1. Introduction

Hepatocellular carcinoma (HCC) ranks as the fourth leading cause of cancer-related deaths in the world [1]. Liver resection is often used as a curative treatment, provided patients do not have underlying cirrhosis in which liver transplantation (LT) remains the only curative option. Liver-directed therapies (LDT) are used to bridge/downstage to LT or as a definitive treatment. Poor response to LDT increases LT waitlist dropout [2] and post-LT recurrence risk [3]. It is important to understand the underlying mechanisms that lead to LDT resistance and HCC progression through noninvasive biomarkers.

Extracellular vesicles are small double-membrane layered structures released from cells into the surrounding environment, which are categorized as exosomes, microvesicles,

or apoptotic vesicles playing an essential role in intercellular communication [4]. In the last 10 years, circulating exosomes (EX) have been extensively investigated as biomarkers [5] and now are an emerging biomarker for several cancers [6]. EXs have been isolated from blood as a possible diagnostic tool for early cancer diagnosis [7]. EXs can participate in intercellular communication within the tumor microenvironment and have been shown to be involved in many processes of HCC including hepatocarcinogenesis, treatment resistance, and metastasis [8,9].

EXs size ranged between 30–200 nm [10] with CD9 and CD63 used as common exosomal surface markers [11]. While much of the work on EXs in HCC has focused on microRNAs, little work has centered on measuring EX shedding in these patients. In this study, we evaluated EX shedding in non-resectable HCC bridged to LT with LDT.

## 2. Materials and Methods

### 2.1. Patient Cohort

This prospective study was approved by the Institutional Reviewer Board within the Ochsner Health System (#2016.131.B) and was performed in accordance with the ethical guidelines of the Helsinki Declaration of 1975. Informed consent was obtained from patients participating in the study. Patients with HCC scheduled to undergo first-cycle liver-directed therapy (LDT) were enrolled between May 2018 and December 2021. All patients were diagnosed with HCC either through biopsy or radiographic imaging in accordance with Liver Imaging Reporting and Data System criteria. Decision for liver-directed therapy was determined by the multidisciplinary HCC board based on the following criteria: (i) Child–Pugh 0–1; (ii) Barcelona Clinic Liver Cancer (BCLC) stage A–B; (iii) non-resectable HCC; (iv) without main portal vein thrombus; (v) without extrahepatic metastasis; (vi) total bilirubin < 4 mg/dL; (vii) creatinine concentration < 1.5 mg/dL; and (viii) absence of gross ascites. First-cycle LDT included drug-eluting embolic transarterial chemoembolization (DEE-TACE), 90Yttrium transarterial radioembolization ($^{90}$Y), or percutaneous microwave ablation (MWA). Patients with ablatable index HCC < 3 cm received first-cycle MWA, while non-ablatable HCC received first-cycle $^{90}$Y. Patients with contraindications for both MWA and $^{90}$Y received DEE-TACE.

All LDT procedures were technically successful. Response to first cycle LDT was scored as objective response rate (ORR) [12] and included those with complete or partial responses using Response Evaluation Criteria in Solid Tumors modified for HCC [13] based on follow-up imaging. Post-treatment imaging was performed either using computed tomography or magnetic resonance imaging. Follow-up imaging was 60 days for MWA and $^{90}$Y and 30 days for DEE-TACE. Clinical variables were extracted from the electronic medical record and included: age, gender, cirrhosis etiology, Child–Pugh and Eastern Cooperative Oncology Group scores, BCLC staging, hepatology labs, and HCC burden (Table 1). Decompensation status was determined based on the need for medical intervention for ascites, hepatic encephalopathy, and/or esophageal varices with bleeding.

### 2.2. Primary Outcome

All patients were in the institution's bridge to liver transplantation protocol, underwent LDT, and were monitored until a primary endpoint of disease progression. The multidisciplinary HCC board determined disease progression, which was defined as progression beyond or failure to downstage to Milan criteria following LDT. Assessment of primary outcome was performed using time to progression (TTP), which was defined as the time from first-cycle LDT until disease progression. The following conditions were censored for TTP analysis: liver transplantation, systemic therapy without tumor progression, lost to follow-up, all-cause mortality, and no evidence of HCC progression at the time of data analysis. Censoring data was defined as the time of the censoring event with the exception of no evidence of HCC progression, which had a censoring date of 31 January 2022.

**Table 1.** Study cohort demographics.

| Demographic | Cohort |
|---|---|
| Patients, n (%) | 43 (100) |
| Age at diagnosis, years (IQR) | 61 (58–68) |
| Sex, male (%) | 33 (76) |
| Race, n (%) | |
| Caucasian | 25 (58) |
| African American | 12 (28) |
| Other | 6 (14) |
| Cirrhotic Etiology, n (%) | |
| HCV | 20 (47) |
| NASH | 10 (23) |
| Other | 13 (30) |
| Cirrhosis Status at Diagnosis, n (%) | |
| Compensated | 35 (81) |
| Decompensated | 8 (19) |
| Scores and Staging | |
| ECOG Performance Status of 0, n (%) | 33 (77) |
| Child–Pugh of A, n (%) | 39 (91) |
| BCLC Stage A, n (%) | 39 (91) |
| **Clinical Hepatology Labs** | |
| Sodium, mM (IQR) | 138 (137–141) |
| Creatinine, mg/dL (IQR) | 0.9 (0.8–1.1) |
| Bilirubin, mg/dL (IQR) | 0.9 (0.5–1.2) |
| Albumin, g/dL (IQR) | 3.5 (3.1–3.8) |
| INR, ratio (IQR) | 1.1 (1.0–1.2) |
| MELD-Na, score (IQR) | 8 (7–10) |
| **Tumour Burden and Biomarkers** | |
| Largest lesion, cm (IQR) | 2.7 (2.2–4.2) |
| Cumulative lesion, cm (IQR) | 3.4 (2.4–4.7) |
| Milan, within criteria (%) | 38 (88) |
| AFP, ng/mL (IQR) | 15 (6.6–44) |
| **First-Line Liver-Directed Therapy** | |
| DEE-TACE, (%) | 4 (9) |
| $^{90}$Y, n (%) | 23 (53) |
| MWA, n (%) | 16 (37) |
| **Treatment Response to First-Line LDT** | |
| ORR | 34 (79) |
| Non-ORR | 9 (21) |
| **Study Endpoint** | |
| Active, n (%) | 21 (49) |
| Tumor progression, n (%) | 8 (19) |
| Transplanted, n (%) | 14 (32) |

Abbreviations: interquartile range (IQR), Hepatitis C virus (HCV), nonalcoholic steatohepatitis (NASH), Eastern Cooperative Oncology Group (ECOG), international normalized ratio (INR), Model for End-Stage Liver Disease-Sodium (MELD-Na), Barcelona Clinic Liver Cancer (BCLC), alpha-fetoprotein (AFP), Doxorubicin-eluting embolic transarterial chemoembolization (DEE-TACE), Yttrium-90 ($^{90}$Y), microwave ablation (MWA); Objective Response Rate (ORR).

### 2.3. Blood Collection

A total of two blood collections occurred from enrolled patients. The first collection was immediately prior to LDT and upon scheduled follow-up imaging post-LDT, the second sample of peripheral blood was collected. All blood was collected in BD Vacutainer® CPT™ mononuclear cell preparation tubes with sodium citrate. Blood was processed according to the manufacturer's protocol to isolate undiluted plasma. All plasma was stored at −80 °C.

### 2.4. Exosome Isolation and Identification

Exosomes were extracted from 500 μL of plasma and isolated through a series of differential centrifugations of supernatants aimed to remove cells and large extracellular vesicles according to a previously published protocol [14] with some modifications. Briefly, plasma was spun at 2000× *g* for 10 min at 4 °C. Supernatant was transferred and spun at 10,000× *g* for 30 min at 4 °C. Supernatant was transferred and centrifuged at 120,000× *g* for 1 h and 30 min using a Beckman Coulter Airfuge air-driven ultracentrifuge with rotor A-95. Exosomes were resuspended in 300–400 μL of 0.03 μm filtered 1× phosphate-buffered saline (PBS).

NanoSight300 Nanoparticle Tracking Analysis (NTA) (NanoSight NS300; Malvern Panalytical, Worestershire, United Kingdom) was used to determine exosome size according to the manufacturer's protocol. For NTA analysis, exosomes were isolated as described previously, but were ultracentrifuged using Beckman Coulter Optima XPN-100 ultracentrifuge with rotor SW41Ti at 100,000× *g* for 2 h at 4 °C. Exosomes were isolated from the HCC cell line Hep3B (ATCC: HB-8064) or plasma from treatment naïve HCC patients. Briefly, Hep3B was grown to 90% confluency as a monolayer in Dulbecco's modified eagle medium (Gibco) supplemented with 10% exosome-depleted fetal bovine serum (ThermoFisher Scientific, Waltham, MA, USA) and 1% antibiotic–antimycotic (ThermoFisher Scientific). Exosomes were isolated from 10 mL of media from the culture. Exosomes were isolated from 2 mL of undiluted plasma from HCC patients. All isolated exosomes were diluted 10 fold with a solution of 0.03 μm filtered 1× PBS supplemented with 2 mM ethylenediaminetetraacetic acid to prevent aggregation of exosomes.

### 2.5. Imaging Flow Cytometry Parameters

Isolated exosomes were analyzed on an ImageStreamX MkII instrument equipped with 5 lasers (405 nm–120 mW, 488 nm–100 mW, 561 nm–150 mW, 642 nm–70 mW, 785 nm–70 mW). The following changes were made to instrument settings: (i) all lasers were set to maximum except 785–10 mW, (ii) magnification was set to 60×, and (iii) 405 nm side scatter (SSC) was enabled. Lasers 405 nm and 785 nm were used for SSC. Protocol for analyzing exosomes was modeled after a recommendation from Luminex Corporation. The core was set to 7 mm and events acquisition to slow. Acquisition time was set to 3 min and volume for each exosome patient sample was recorded.

The following gating strategy was developed as shown in Figure 1 Briefly, Speed-Beads were gated out and ApogeeMIX calibration beads (Apogee Flow Systems, Heftfordshire, United Kingdom, Cat#1493) were used to determine the size gate for exosomes on 405 nm and 785 nm SSC. The following antibodies were used: Alexa Fluor 647 (AF647)-conjugated anti-cluster of differentiation (CD) 9 (ThermoFisher Scientific, clone MEM-61), PE-Cy7-conjugated anti-CD63 (ThermoFisher Scientific, clone H5C6), PE-Cy7-conjugated anti-mouse IgG1k isotype control (ThermoFisher Scientific, clone P3.6.2.8.1), and AF647-conjugated anti-mouse IgG2a kappa isotype control (ThermoFisher Scientific, clone eBM2a). Isotype and antibody-only controls were used to determine the level of background. Samples were stained for 1 h before acquisition. Some isolated exosomes required dilution with 0.03 μm filtered 1× PBS prior to analysis on the ImageStreamX MkII instrument. Data analysis was performed using Amnis IDEAS software (version 6.2.187.0, Fremont, CA, USA) and Beckman Coulter Kaluza.

### 2.6. Statistical Analysis

Data analysis was performed using JMP 13.0 (SAS Institute Inc., Cary, NC, USA) with graphical output generated using GraphPad Prism 8.4.3 (GraphPad Software Inc., San Diego, CA, USA). Categorical variables are shown as a number and percentage of the cohort unless otherwise stated. All continuous variables were displayed as median with interquartile (IQR) range. Matched paired analysis was used to determine differences between baseline and post-treatment exosome shedding using the Wilcoxon Signed Rank test. Cochran Armitage Trend test was used to determine significant associations between

categorical variables. Logistic regression analysis was used for factors associated with exosome shedding after treatment. Kaplan–Meier survival curves were generated in GraphPad Prism 8.4.3 and compared using log-rank tests.

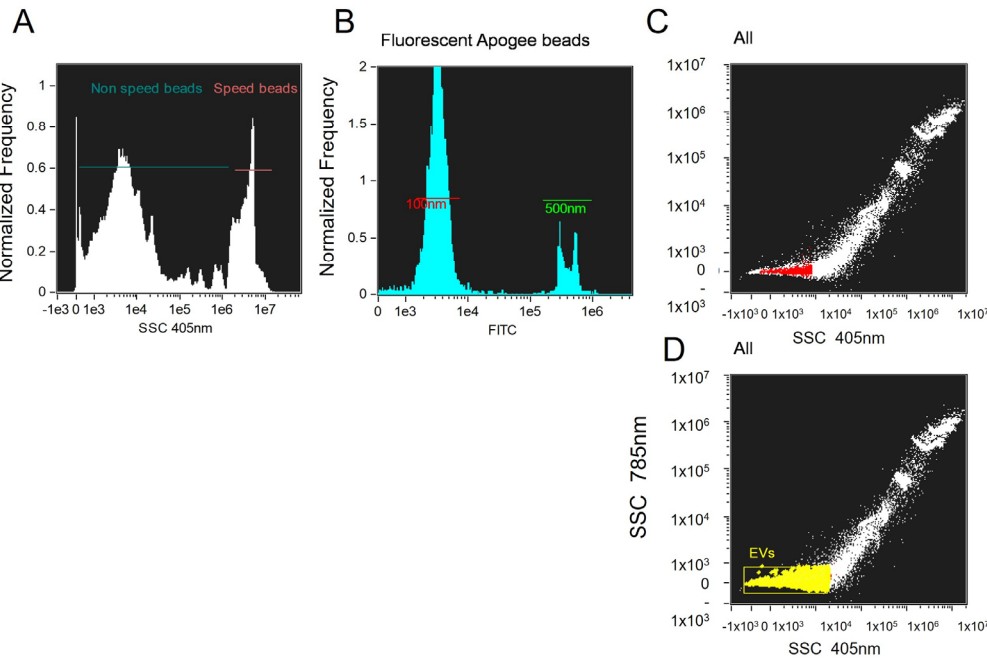

**Figure 1.** Gating strategy for purified exosomes on Amnis ImageStream. Two methods were used to remove speed beads based on (**A**) side scatter at 405 nm and (**B**) FITC intensity. A size selection was performed using SSC at 785 nm by 405 nm. (**C**) Located in red are the selected events based on size. (**D**) Final gate for exosomes.

## 3. Results

### 3.1. Protocol Setup for Detection of EXs Using Imaging Flow Cytometry

Ultracentrifugation was used to isolate exosomes from both the HCC cell line Hep3B and plasma from two HCC patients. Exosomes were resuspended in 0.03 μm filtered PBS. NTA confirmed undetectable particles in filtered PBS (Figure 2A). NTA also showed differences in exosome diameter between the Hep3B-derived exosomes and HCC patient-derived exosomes. Hep3B-derived exosomes had a mean diameter of 139 nm (Figure 2B), while HCC patient-derived exosomes displayed a larger mean exosome diameter of 142 nm and 195 nm, respectively (Figure 2C,D).

Imaging flow cytometry (IFC) was used to detect and quantify exosomes from treatment-naïve HCC patients before and after liver-directed therapy. Several controls were tested prior to establishing a final protocol for the detection and quantification of exosomes for patient samples. Amnis ImagestreamX MkII uses SpeedBeads for calibration of the instrument and required removal based on side scatter (Figure 1A). After SpeedBead removal, ApogeeMIX calibration beads, a mixture of silica and polystyrene FITC-labeled beads ranging in size from 100–500 nm, were used to set size-specific gates for exosomes (Figure 1B). A buffer control (0.03 μm filtered PBS), antibody-only controls (0.03 μm filtered PBS + antibody), and isotype controls were used to determine the level of background. Although low, antibody-only and isotype controls produced antibody-mediated background while low background was observed in the buffer control (Figure 3A–C).

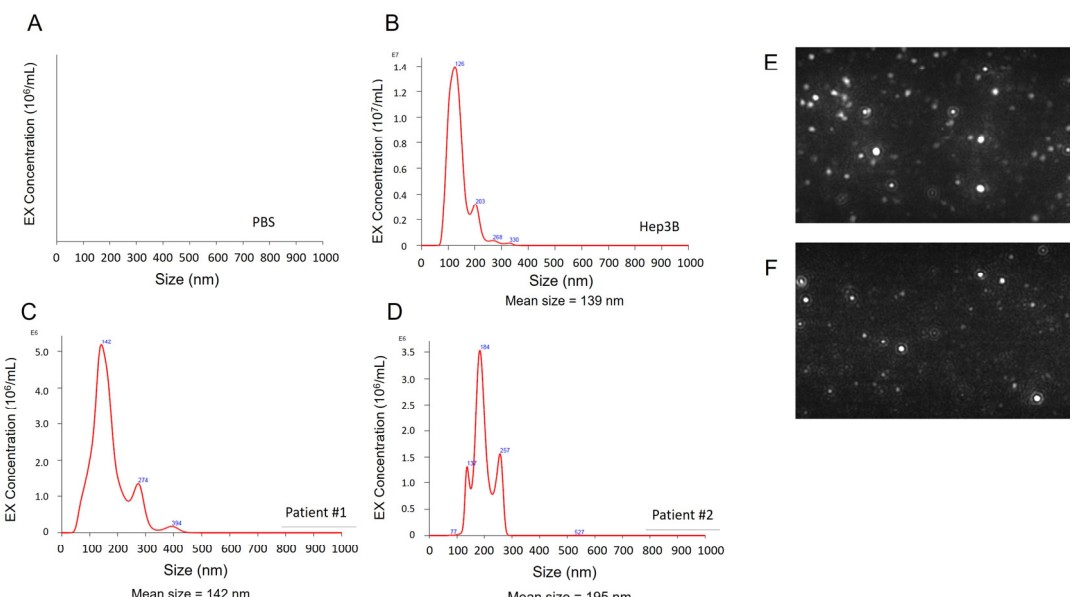

**Figure 2.** Confirmation of purified exosomes with NanoSight Nanoparticle Tracking Analysis. Purified exosomes were isolated and analyzed using NanoSight. EX concentration for (**A**) 0.03 μm filtered PBS, (**B**) HCC cell line Hep3B, and representatives from two patients with HCC (**C**,**D**). Actual view from NTA of purified exosomes from (**E**) Hep3B and (**F**) representative patient. Abbreviations: Hepatocellular carcinoma (HCC), phosphate-buffered saline (PBS), nanotracking analysis (NTA), exosomes (EXs).

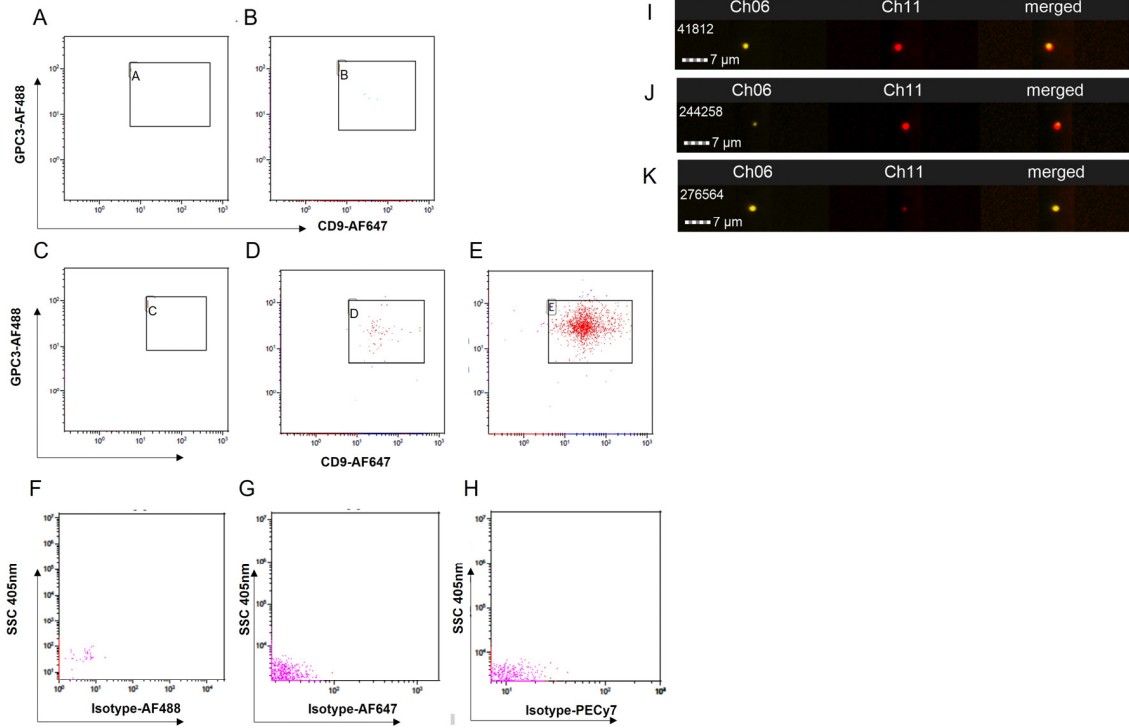

**Figure 3.** Enumeration and visualization of patient-purified exosomes using flow cytometry. After gating (**A**) 0.03 μm filtered PBS, (**B**) representative antibody-only control, (**C**) representative unstained patient-derived exosomes, representative patient-derived exosomes at (**D**) baseline, and (**E**) post treatment. Isotype controls for (**F**) AF488, (**G**) AF647, and (**H**) PECy7. Visual confirmation of representatives purified exosomes from HCC patient (**I**–**K**). Abbreviations: phosphate-buffered saline (PBS), hepatocellular carcinoma (HCC).

### 3.2. Patient Demographics

The cohort consisted of 43 patients with non-resectable HCC undergoing first-cycle liver-directed therapy (Table 1). The cohort had a median age of 61 years, mostly male (33/43, 76%) with Hepatitis C (HCV) as the predominant etiology of cirrhosis (20/43, 47%). Overall, patients had well-compensated liver disease (35/43, 81%) with a median Model for End-stage Liver Disease-sodium (MELD-Na) of 8 and Child–Pugh A (39/43, 91%). Patients were mostly BCLC stage A (39/43, 91%) with median index lesion size of 2.7 cm and median alpha-fetoprotein of 15 ng/mL. Most patients underwent $^{90}$Y as first-cycle treatment (23/43, 53%), followed by MWA (16/43, 37%) and DEE-TACE (4/43, 9%). An ORR to first-cycle LDT was achieved in 79% of patients (34/43). In patients that reached the primary endpoint, 36% (8/22) experienced disease progression, while 64% (14/22) received a liver transplant. The remaining cohort (21/43, 49%) remained active under transplant evaluation.

### 3.3. EX Shedding Following LDT

Two exosomal markers, CD9 and CD63, were used to detect and quantify EXs in treatment-naïve HCC patients at baseline and following first-cycle LDT using imaging flow cytometry. Differential expression of exosomal markers was observed with 10 fold more EXs expressing CD9$^+$ compared to CD63$^+$ (Table 2). Although LDT increased exosome shedding for both CD9$^+$ and CD63$^+$ expressing EXs, this failed to reach significance. Grouped matched pairs analysis revealed total EXs shedding after LDT was independent of response to the first-cycle response and primary endpoint (Supplemental Table S1). Patients were divided into two groups, high and low, based on median split of CD9$^+$ and CD63$^+$ EXs post-LDT, respectively (Supplemental Figure S1). Overall, there were more patients that had high EX shedding post-LDT for both CD9$^+$ (low, 16 patients to high, 20 patients) and CD63$^+$ (low, 17 patients to high, 20 patients) (Table 2).

**Table 2.** Exosome shedding after LDT in HCC.

| EXosome Shedding | Baseline | Post-LDT | *p* Value |
|---|---|---|---|
| **EXs by marker, median (IQR)** | | | |
| CD9$^+$ | $7.2 \times 10^6$ ($3.8 \times 10^6$–$17.4 \times 10^6$) | $10.7 \times 10^6$ ($4.3 \times 10^6$–$20.7 \times 10^6$) | 0.545 |
| CD63$^+$ | $0.5 \times 10^6$ ($0.3 \times 10^6$–$1.1 \times 10^6$) | $0.7 \times 10^6$ ($0.4 \times 10^6$–$0.9 \times 10^6$) | 0.734 |
| Double positive, CD9$^+$ CD63$^+$ | $0.08 \times 10^6$ ($0.03 \times 10^6$–$0.2 \times 10^6$) | $0.1 \times 10^6$ ($0.05 \times 10^6$–$0.4 \times 10^6$) | 0.632 |
| **Exosome Shedding Group** | | | |
| **CD9$^+$ EXs, n (%)** | | | 0.042 |
| High | 16 (41) | 20 (50) | |
| Low | 23 (59) | 20 (50) | |
| **CD63$^+$ EXs, n (%)** | | | 0.004 |
| High | 17 (44) | 20 (50) | |
| Low | 22 (56) | 20 (50) | |
| **Double positive, CD9$^+$ CD63+ EXs, n (%)** | | | 0.015 |
| High | 17 (44) | 20 (50) | |
| Low | 22 (56) | 20 (50) | |

Abbreviations: liver-directed therapy (LDT); Hepatocellular carcinoma (HCC); exosomes (EXs); interquartile range (IQR).

### 3.4. Prognostic Factors Associated with EX Shedding Following LDT

Due to the increased expression of CD9$^+$ versus CD63$^+$ on EXs in HCC patients, CD9$^+$ was used for downstream analysis. We hypothesized a correlation between EX shedding and response to first-cycle LDT. Variables at the diagnostic baseline including demographics or cirrhosis serology were not associated with the degree of EX shedding (Table 3). Variables related to HCC including tumor size and staging revealed only BCLC staging was associated with post-treatment EX shedding level. In patients with high EX shedding, 95% (19/20) had an ORR to first-cycle LDT compared to those in the low EX

shedding group (70%, 14/20). All patients with high CD9$^+$ EX shedding (100%, 10/10) were successfully bridged to liver transplantation compared to 22% (2/9) of patients with low CD9$^+$ shedding who experienced tumor progression.

**Table 3.** Exosome shedding post-LDT and clinical variables.

| Demographic | CD9$^+$ EXs | | |
| --- | --- | --- | --- |
| | **High** | **Low** | *p* **Value** |
| Age at diagnosis, median (IQR) | 61 (58–67) | 62 (55–68) | 0.926 |
| Sex, male, n (%) | 17 (85) | 14 (70) | 0.252 |
| Race, n (%) | | | 0.180 |
| Caucasian | 13 (65) | 10 (50) | |
| African American | 3 (15) | 8 (40) | |
| Other | 4 (20) | 2 (10 | |
| **Cirrhotic etiology, n (%)** | | | 0.066 |
| HCV | 11 (55) | 8 (40) | |
| NASH | 3 (15) | 7 (35) | |
| Other | 6 (30) | 5 (25) | |
| **Cirrhosis status at diagnosis, n (%)** | | | 0.427 |
| Compensated | 15 (75) | 17 (85) | |
| Decompensated | 5 (25) | 3 (15) | |
| **Scores and Staging** | | | |
| ECOG performance status 0, n (%) | 15 (75) | 15 (75) | 1.0 |
| Child–Pugh A, n (%) | 17 (85) | 19 (95) | 0.282 |
| BCLC HCC stage A, n (%) | 20 (100) | 17 (85) | 0.036 |
| **Clinical Hepatology Labs** | | | |
| Sodium | 138 (137–141) | 139 (136–140) | 0.866 |
| Creatinine | 1.0 (0.8–1.1) | 0.9 (0.8–1.2) | 0.219 |
| Bilirubin | 0.7 (0.5–1.0) | 1.1 (0.6–1.3) | 0.181 |
| Albumin | 3.8 (3.1–4.1) | 3.4 (3.1–3.6) | 0.901 |
| INR | 1.1 (1.0–1.3) | 1.1 (1.0–1.2) | 0.184 |
| MELD-Na | 8 (7–10) | 9 (7–11) | 0.527 |
| **Tumor Burden and Biomarkers** | | | |
| Largest lesion | 2.7 (2.3–3.6) | 3.1 (2.2–4.8) | 0.187 |
| Cumulative lesion | 3.0 (2.3–4.7) | 3.8 (2.6–4.8) | 0.233 |
| Milan criteria | 19 (95) | 17 (85) | 0.282 |
| AFP | 7.8 (4.0–38) | 21 (8.2–75) | 0.067 |
| **First-Line Liver-Directed Therapy** | | | 0.066 |
| DEE-TACE, n (%) | 2 (10) | 0 (0) | |
| $^{90}$Y, n (%) | 8 (40) | 14 (70) | |
| MWA, n (%) | 10 (50) | 6 (30) | |
| **Treatment Response to First-Cycle LDT** | | | 0.030 |
| ORR, n (%) | 19 (95) | 14 (70) | |
| Non-ORR, n (%) | 1 (5) | 6 (30) | |
| **Study Endpoint** | | | <0.001 |
| Tumor progression, n (%) | 0 (0) | 7 (78) | |
| Transplanted, n (%) | 10 (100) | 2 (22) | |

Abbreviations: exosomes (EXs), interquartile range (IQR), Hepatitis C virus (HCV), nonalcoholic steatohepatitis (NASH), Eastern Cooperative Oncology Group (ECOG), international normalized ratio (INR), Hepatocellular carcinoma (HCC); Model for End-stage Liver Disease-sodium (MELD-Na), Barcelona Clinic Liver Cancer (BCLC), alpha-fetoprotein (AFP), Doxorubicin-eluting embolic transarterial chemoembolization (DEE-TACE), Yttrium-90 ($^{90}$Y), microwave ablation (MWA); liver-directed therapy (LDT); Objective Response Rate (ORR).

### 3.5. Post-LDT EX Shedding and TTP

Overall, TTP following LDT at 1 and 2 years for the cohort was 84% and 65%, respectively (Figure 4A). Patients with ORR showed superior TTP at 1 and 2 years of 96% and 75% compared to those with non-ORR with TTP of 25% and 0%, respectively (Figure 4B). Patients with high CD9+ EX shedding after treatment showed superior TTP at 1 and 2 years. Progression rates were 30% and 65% at 1 and 2 years post-LDT for patients with low CD9+ EX shedding. To test CD9+ EX shedding as prognostic for TTP in those patients with ORR, TTP in only those patients who achieved an ORR after first-cycle LDT was investigated (Figure 4D). CD9+ EX shedding stratified 2-year progression risk in patients with ORR with low CD9+ EX shedding at a progression rate of 55% compared to 0% in the high CD9+ EX shedding group.

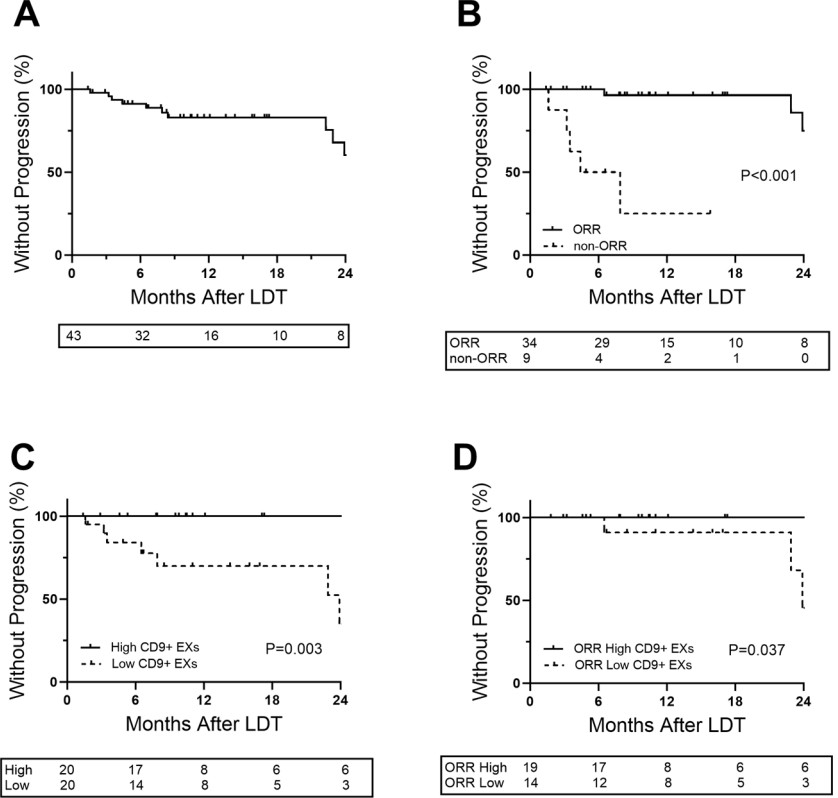

**Figure 4.** Exosome shedding and time to progression. (**A**) Kaplan–Meier curve of overall time to progression. (**B**) Kaplan–Meier curve of time to progression based on first cycle LDT response. (**C**) Kaplan–Meier curve of time to progression based on exosome shedding. (**D**) Exosome shedding in patients with ORR to first cycle LDT. Number of at-risk patients at each time interval is shown in the table. Log-rank test was used to determine significance. Abbreviation: exosomes (EXs), liver-directed therapy (LDT), objective response rate (ORR).

## 4. Discussion

Given their small size (<200 nm), detection and quantification of EXs remains challenging. Confirmation of isolated EXs through ultracentrifugation or sucrose density gradient centrifugation has been achieved by staining exosomal markers using immunoblotting, enzyme-link immunosorbent, and immunohistochemistry. These techniques stain one surface marker at a time, which is time-consuming and tedious. Electron microscopy and nanoparticle tracking analysis have been used for visualization and quantification of EXs, but these methods lack the specificity of staining with exosomal surface markers. IFC to detect and quantify EXs has gained traction in recent years [15,16], with several benefits over previously mentioned methods. First, IFC is extremely sensitive for the detection of EXs compared to immunoblotting or immunohistochemistry. Second, multiple surface

exosomal markers can be used simultaneously with IFC. Third, like NTA, both numeration and visualization of EXs can be performed. IFC combines the benefits of NTA with the specificity of immunoblotting and immunohistochemistry together. Taken together, IFC is extremely sensitive and allows for the positive selection of multiple EX markers, visualization, and numeration of EXs in a fast throughput manner. In this study, an IFC protocol specific for EX detection and quantification [16] was modified for the detection of isolated EXs from early-stage HCC patients. This modified protocol was shown to be effective for quantifying EX shedding before and after LDT, providing a successful application of IFC in detecting EX shedding in HCC patients.

There is limited research into the prognostic value of EX shedding and the impact of liver-directed therapy on EX shedding in non-resectable HCC. Research has largely focused on EX cargo and their impact on HCC progression and metastatic potential [17,18]; see reviews [19,20] focusing on EXs isolated from HCC cell lines [21]. Those studies isolating EXs from the serum/plasma of HCC patients have honed in on EX cargo [22,23], ignoring shedding. In addition to cargo, expression levels of EX-specific markers are not universal across all cancers and may be dependent on EX size [24] or where EXs form [25]. Studies investigating EXs in a number of diverse cancers have also shown differential expression of exosomal markers through immunoblotting [26] and immunohistochemistry [26]. In this study, EX shedding was monitored in early-stage HCC patients before and after LDT using IFC which allowed for quantification of two EX-specific markers. Isolated EXs from HCC patients had an average size of <200 nm, in line with previous studies [18,22,23]. CD9 was expressed more frequently on EXs than CD63 in non-resectable HCC patients. While differential expression between CD63$^+$ and CD9$^+$ was also found on EXs from patients with chronic hepatitis B and acute-on-chronic liver failure [15], how this difference impacts response to treatment, EX shedding, or progression rates in HCC patients remains unknown. These results suggest that when quantifying EXs, multiple surface markers should be used to account for expression differences in different populations.

Clinical applications of EX shedding for HCC have focused on understanding the metastatic potential based on exosomal cargo for diagnosis or as potential therapeutic targets (see review [27]). In this study, total EXs were quantified regardless of origin as a prognostic factor for TTP risk after treatment with LDT as a bridge to LT. Exosomal cargo changes on the microRNA [28] or protein level [29] following LDT have been shown to be associated with survival [28] or response to treatment [29]. In this study, low EX shedding was associated with an increased risk of progression following treatment. Even in patients with ORR, low EX shedding resulted in significantly lower TTP. A response to LDT relies on radiographic imaging and measurements of alpha-fetoprotein. In patients with high EX shedding, 95% experienced ORR to LDT compared to just 70% in the low shedding group. Low EX shedding may indicate resistance to LDT with viable tumors remaining, while those patients with high EX shedding could result from increased tumor cell death after treatment. Most patients require multiple treatments of LDT while being bridged to LT. In a large multi-centered study, 59% of patients required multiple LDTs, with 31% of the cohort requiring ≥ 3 LDTs until the endpoint [2]. In addition to that, 82% of patients with complete responses to LDT have viable tumors at explant [2]. In patients with low EX shedding that reached the endpoint, 56% (5/9) received additional treatment. Despite this, 78% of patients with low EX shedding that reached the endpoint experienced disease progression precluding access to LT compared to 0% in those patients with high EX shedding. EX shedding may be a useful mechanism to monitor treatment of HCC with destruction of the tumor leading to increased EX shedding.

In conclusion, this study provides a new approach for measuring EX shedding in HCC patients that could be used to predict patients are risk of HCC progression.

**Supplementary Materials:** The following supporting information can be downloaded at: https://www.mdpi.com/article/10.3390/livers3040047/s1, Table S1: exosome shedding by first cycle response to LDT in HCC; Figure S1: baseline and post-treatment exosome shedding.

**Author Contributions:** Conceptualization, K.G.N., D.W. and P.T.T.; methodology, K.G.N., D.W. and P.T.T.; software, K.G.N., D.W. and P.T.T.; formal analysis, K.G.N., D.W., T.S. and P.T.T.; investigation, K.G.N., D.W. and M.H.; resources, S.D., A.J.C. and P.T.T.; data curation, K.G.N., D.W., M.H. and T.S.; writing—original draft preparation, K.G.N., D.W. and P.T.T.; writing—review and editing, K.G.N., D.W., M.H., T.S., J.G., A.R.K., Y.A., S.D., A.J.C. and P.T.T.; visualization, K.G.N., D.W. and P.T.T.; supervision, A.J.C. and P.T.T.; project administration, P.T.T.; funding acquisition, S.D., A.J.C. and P.T.T. All authors have read and agreed to the published version of the manuscript.

**Funding:** This research was funded by the American Society of Transplant Surgeons Collaborative Scientist Grant.

**Institutional Review Board Statement:** This study was conducted in accordance with the Declaration of Helsinki and approved by the Institutional Review Board of Ochsner Health System (protocol 2016.131.B).

**Informed Consent Statement:** Informed consent was obtained from all subjects involved in the study.

**Data Availability Statement:** The dataset generated for the current study is available from the corresponding author upon reasonable request and IRB-mandated approval of a data use agreement.

**Conflicts of Interest:** The authors declare no conflict of interest.

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
