# Peer review of "Exosome Shedding Is Concordant with Objective Treatment Response Rate and Stratifies Time to Progression in Treatment Naïve, Non-Resectable Hepatocellular Carcinoma"

_livers, doi:10.3390/livers3040047_

Round 1
Reviewer 1 Report
Comments and Suggestions for Authors
Abstract:
The authors state exosome (EV) and it is suggested to abbreviate it correctly: The first sentence to read as: Translational strategies to characterize and monitor Extracellular Vesicles (EV) such as exosome shedding ......
The second sentence to read as: Purified EVs were stained using markers CD9 and CD63
2.3 Blood Collection: Prior to LDT if authors collected PBMCs or plasma and what time points for the first and subsequent collection were not specified.
Comments on the Quality of English LanguageSpell check, sentence construction, etc., are to be reviewed prior to submission of the final version.
Author Response
Comments and Suggestions for Authors
Abstract:
The authors state exosome (EV) and it is suggested to abbreviate it correctly: The first sentence to read as: Translational strategies to characterize and monitor Extracellular Vesicles (EV) such as exosome shedding ......
We apologize for this oversight and recognize our abbreviation for exosomes was confusing. We have updated all abbreviations within the manuscript text and figures for exosomes to be EX instead of ‘EV’
The second sentence to read as: Purified EVs were stained using markers CD9 and CD63
We thank the reviewer and have fixed this sentence in the text (line 29).
2.3 Blood Collection: Prior to LDT if authors collected PBMCs or plasma and what time points for the first and subsequent collection were not specified.
We apologize for this oversight. We have clarified this section to better explain the time points for blood collections (lines 107-112).
Comments on the Quality of English Language
Spell check, sentence construction, etc., are to be reviewed prior to submission of the final version.
The manuscript was reviewed again for spelling and sentence structure errors.
Reviewer 2 Report
Comments and Suggestions for Authors
Thank you for the opportunity to review this very nice manuscript. The authors address the need for better prognostic tools for treatment modalities of early-stage HCC in terms of time to disease progression following liver-directed therapy (LDT). They quantify circulating exosomes post LDT in a cohort of 43 patients with non-resectable early-stage HCC undergoing first cycle LDT and find that high exosome shedding correlated with successful bridge to liver transplantation. The paper is well-written and the data is clearly presented. I envision some difficulty in currently making this a widely used prognostic biomarker, but as technology progresses, these methods will become more readily available.
Major critique:
The study is limited as a small single center study, but provides promising data that supports advancement to a larger multi-center study.
Technically, the measuring of exosomes will be limiting at most institutions will not have access to the necessary ultracentrifugation or imaging flow cytometry facilities.
An hypothesis of whey exosome shedding is a good prognostic occurrence after LDT should be added. Perhaps the exosomes are shedded in response to tumor cell death brought about by the therapy or by the subsequent development of a cellular immune response? Unclear, but speculation is welcome.
Minor points.
Fig. 1, Y-axis labels need to be larger. Same with axis labels in Fig. 2.
Fig. 3, has antibody only control by really needs an isotope control as exosomes may carry CD16/32 (FcR) on their surface.
Author Response
Comments and Suggestions for Authors
Thank you for the opportunity to review this very nice manuscript. The authors address the need for better prognostic tools for treatment modalities of early-stage HCC in terms of time to disease progression following liver-directed therapy (LDT). They quantify circulating exosomes post LDT in a cohort of 43 patients with non-resectable early-stage HCC undergoing first cycle LDT and find that high exosome shedding correlated with successful bridge to liver transplantation. The paper is well-written and the data is clearly presented. I envision some difficulty in currently making this a widely used prognostic biomarker, but as technology progresses, these methods will become more readily available.
Major critique:
The study is limited as a small single center study, but provides promising data that supports advancement to a larger multi-center study.
We agree with the reviewer. We hope to expand this research to include additional sites to provide a larger multi-center study.
Technically, the measuring of exosomes will be limiting at most institutions will not have access to the necessary ultracentrifugation or imaging flow cytometry facilities.
Again, we agree with the reviewer. Most institutions will not have access to the instruments used in this study. With this manuscript, we are open the door to new techniques for exosomal research.
An hypothesis of whey exosome shedding is a good prognostic occurrence after LDT should be added. Perhaps the exosomes are shedded in response to tumor cell death brought about by the therapy or by the subsequent development of a cellular immune response? Unclear, but speculation is welcome.
We thank the reviewer for the suggestion. We do provide some speculation regarding low exosome shedding following treatment possibly due to resistance to treatment (line 349). We have provided addition text to expand upon this idea in the conclusion (line 350).
Minor points.
Fig. 1, Y-axis labels need to be larger. Same with axis labels in Fig. 2.
We thank the reviewer for this suggestion. We have increased the font on the axis for both Figure 1 and Figure 2.
Fig. 3, has antibody only control by really needs an isotope control as exosomes may carry CD16/32 (FcR) on their surface.
We thank the reviewer for the comment. Isotype controls were used to set up initial gating strategy The background was extremely low. We have updated Figure 3 to include the isotype controls and have added this information into the text (lines 150-152; 188-189).